# Products of Docosahexaenoate Oxidation as Contributors to Photosensitising Properties of Retinal Lipofuscin

**DOI:** 10.3390/ijms22073525

**Published:** 2021-03-29

**Authors:** Małgorzata B. Różanowska, Anna Pawlak, Bartosz Różanowski

**Affiliations:** 1School of Optometry and Vision Sciences, Cardiff University, Cardiff CF24 4HQ, Wales, UK; 2Cardiff Institute for Tissue Engineering and Repair (CITER), Cardiff University, Cardiff CF24 4HQ, Wales, UK; 3Department of Biophysics, Faculty of Biochemistry, Biophysics and Biotechnology, Jagiellonian University, 30-387 Kraków, Poland; Anna.Pawlak@uj.edu.pl; 4Institute of Biology, Pedagogical University, 30-084 Kraków, Poland; Bartosz.Rozanowski@up.krakow.pl

**Keywords:** retina, retinal pigment epithelium, lipofuscin, docosahexaenoate, docosahexaenoic acid, photosensitiser, photosensitised oxidation, singlet oxygen, superoxide, reactive oxygen species

## Abstract

Retinal lipofuscin which accumulates with age in the retinal pigment epithelium (RPE) is subjected to daily exposures to high fluxes of visible light and exhibits potent photosensitising properties; however, the molecules responsible for its photoreactivity remain unknown. Here, we demonstrate that autooxidation of docosahexaenoate (DHE) leads to the formation of products absorbing, in addition to UVB and UVA light, also visible light. The products of DHE oxidation exhibit potent photosensitising properties similar to photosensitising properties of lipofuscin, including generation of an excited triplet state with similar characteristics as the lipofuscin triplet state, and photosensitised formation of singlet oxygen and superoxide. The quantum yields of singlet oxygen and superoxide generation by oxidised DHE photoexcited with visible light are 2.4- and 3.6-fold higher, respectively, than for lipofuscin, which is consistent with the fact that lipofuscin contains some chromophores which do contribute to the absorption of light but not so much to its photosensitising properties. Importantly, the wavelength dependence of photooxidation induced by DHE oxidation products normalised to equal numbers of incident photons is also similar to that of lipofuscin—it steeply increases with decreasing wavelength. Altogether, our results demonstrate that products of DHE oxidation include potent photosensitiser(s) which are likely to contribute to lipofuscin photoreactivity.

## 1. Introduction

Retinal lipofuscin accumulates with age in the retinal pigment epithelium (RPE) cells in the form of cytoplasmic inclusion bodies occupying almost 20% of cytoplasmic volume by the age of 80 years and can be monitored via its characteristic golden-yellow fluorescence which is used as a diagnostic tool in various retinal diseases [1,2,3,4,5]. It has been shown that retinal lipofuscin exhibits potent photosensitising properties—when photoexcited under aerobic conditions, it generates singlet oxygen and superoxide, oxidises lipids and proteins and is responsible for age-related increase in RPE susceptibility to photooxidation and phototoxicity [6,7,8,9,10,11,12,13,14,15,16]. To date, several visible-light-absorbing chromophores have been identified in the RPE lipofuscin which are derived from interactions of two molecules of vitamin A aldehyde with phosphatidylethanolamine, forming various bisretinoids and their oxidation products [7,17,18,19,20,21,22,23,24]. However, the photosensitising properties of these bisretinoids cannot explain the photosensitising properties of lipofuscin (reviewed in [25,26,27]). 

In particular, it has been demonstrated that the bisretinoids exhibit characteristic absorption spectra with two absorption maxima-one in the ultraviolet and the other in the visible spectral range, and a local minimum in between these peaks [24]. The wavelength dependence of photooxidation of the most widely studied bisretinoid, known as A2-E, matches well with the absorption spectrum of A2-E, whereas the wavelength dependence of photooxidation—for both whole lipofuscin granules and the lipophilic extract of lipofuscin—exhibit a monotonic increase with decreasing wavelength consistent with the absorption spectrum of the lipofuscin extract [6,12,15]. 

Moreover, it has been demonstrated that photoexcitation of A2-E leads to the formation of an excited triplet state, and the superoxide and singlet oxygen with substantially lower quantum yields than the corresponding quantum yields for the lipophilic extract of lipofuscin [7,14,28,29,30,31,32]. In particular, the quantum yield of superoxide is 3.2 times greater for the lipophilic extract of lipofuscin than for A2-E [14]. The reported values of the quantum yield of singlet oxygen generation resulting from photoexcitation of A2-E with ultraviolet or blue light (420 nm) vary from 0.0008 in polar deuterated ethanol to 0.004 in non-polar hexafluorobenzene [32]. For a lipophilic extract of lipofuscin solubilised in benzene, the quantum yields of singlet oxygen generation are at least 12-fold greater than for A2-E, 0.08 and 0.05, respectively, when excited with 355 nm or blue light (420, 430 or 440 nm), and they increase to 0.15 and 0.09, respectively, upon saturation with oxygen [9]. This clearly indicates that lipofuscin includes photosensitisers, which are more potent than A2-E. However, it remains unknown which molecule(s) are responsible for lipofuscin photosensitising properties.

RPE lipofuscin includes high concentrations of polyunsaturated lipids [33]. Modifications of proteins by-products of lipid oxidation have been identified in proteins associated with lipofuscin and have been suggested to be involved in both the formation of lipofuscin and its deleterious effects [21,34,35,36,37,38,39]. It has been reported that, when photoexcited with UVB or UVA light, oxidised polyunsaturated fatty acids, such as linoleic, linolenic, arachidonic acid, and adducts with protein/lysine of lipid oxidation products, such as malondialdehyde, can photogenerate singlet oxygen [40,41,42,43]. However, no products of lipid oxidation were reported, which can absorb visible light and/or photosensitise singlet oxygen generation upon photoexcitation with visible light as lipofuscin does. 

While working with phosphatidylcholines containing docosahexaenoate (DHE) as their acyl chain(s), we noticed that, when the lipid powder is exposed to the air, it turns yellow, indicating that it may absorb blue light. DHE is the most unsaturated fatty acid identified in RPE lipofuscin, accounting for 6.1–9.1% and 7.2–16.0% of fatty acids in phospholipids (mainly phosphatidylcholine) and free fatty acids, respectively [33]. DHE is essential for the proper function of the retina [44,45,46,47,48,49,50,51,52]. However, due to the presence of six unsaturated double bonds, DHE is very susceptible to non-enzymatic oxidation, forming a large number of various oxidation products [39,53,54,55,56,57,58,59]. Products of DHE oxidation have been detected in RPE lipofuscin isolated from human cadaver eyes, namely, carboxyethylpyrrole (CEP) [21]. CEP can be formed as a result of adduction of DHE oxidation product, 4-hydroxy-7-oxohept-5-enoic acid (HOHA) lactone to primary amine groups of lysine residues on proteins or ethanolamine groups of phospholipids [60,61]. The HOHA lactone is formed due to the spontaneous deacylation of HOHA-containing phosphatidylcholine (HOHA–PC) [62]. HOHA–PC has been identified in the albino rodent neural retina as a result of exposure to green light, which is absorbed by visual pigments, demonstrating that oxidation of DHE can occur readily in vivo [63]. Despite the fact that many DHE oxidation products were identified, none of the reported structures of these oxidation products has a sufficiently delocalised electron to be able to absorb the visible light.

The purpose of this study was to evaluate the possibility that oxidation of DHE can lead to the formation of products absorbing visible light and having similar photosensitising properties as RPE lipofuscin. To test this hypothesis, we oxidised DHE by exposure to the air and compared its photochemical properties with RPE lipofuscin. Our results demonstrate that autooxidation of DHE leads to the formation of products absorbing in addition to UVB and UVA light, also visible light. The absorption spectrum of oxidised DHE shows a similar characteristic increase in absorbance with decreasing wavelength as that observed for the lipophilic extract of lipofuscin. We have demonstrated that oxidation products of DHE exhibit potent photosensitising properties similar to photosensitising properties of lipofuscin, including the generation of an excited triplet state with similar characteristics as the lipofuscin triplet state, and photosensitised formation of singlet oxygen and superoxide. The quantum yields of singlet oxygen and superoxide generated by oxidised DHE photoexcited by visible light are 2.4- and 3.6-fold higher, respectively, than those reported for lipofuscin. However, it is already known that lipofuscin contains some chromophores, which do contribute to the absorption of light but not so much to its photosensitising properties. Importantly, the wavelength dependence of photooxidation induced by DHE oxidation products normalised to equal numbers of incident photons is also similar to that of lipofuscin—it steeply increases with decreasing wavelength. Altogether, our results demonstrate that oxidised DHA is a potent photosensitiser, which is likely to contribute to lipofuscin photoreactivity and thereby its phototoxicity.

## 2. Results

### 2.1. Formation of Products Absorbing near Ultraviolet and Visible Light during Autooxidation of Docosahexaenoic Acid (DHA) and Docosahexaenoate (DHE) Phospholipids

Exposure to the air of docosahexaenoic acid (DHA) oil led to a decrease in the absorption peak of DHA and an increase in oxidation products with absorption maxima at about 235 nm (Figure 1A). Monitoring absorption spectra at a 10-fold greater concentration revealed a shoulder at about 270–280 nm and a tail extending towards longer wavelengths (Figure 1B). After three days of autoxidation, the absorption of oxidation products monitored at a concentration of 100 mg/mL extended above 390 nm, that is, into the visible range of the electromagnetic spectrum (Figure 1C). With further oxidation time, the absorption extended further into longer wavelengths and after 28 days extended up to 550 nm.

Similar changes in absorption were observed upon exposure to the air for 28 days of phosphatidylcholine containing either one or two docosahexaenoyl (DHE) acyl chains (Figure 1D). The absorbance increase with decreasing wavelength of oxidised DHE-containing phospholipids was considerably less steep than the absorbance spectrum of oxidised DHA (Figure 1C,D)—a comparison of DHA and DHE oxidised for 28 days shows that when they both reached the same absorbance value of 0.028 at 548 nm, their absorbance values of 1 were at 440 nm and 380 nm, respectively, for DHA and DHE. Clearly, absorption of DHE appears to fit better into the absorption spectrum of lipophilic extract of lipofuscin than absorption of DHA (Figure 1E).

The DHE-containing phospholipids remained soluble in organic solvents after oxidation and, in the case of didocosahexaenoyl-phosphatidylcholine, it became soluble even in the aqueous solution. In contrast, the extensive oxidation of DHA oil led to products of decreased solubility. Due to the spectral characteristics and solubility, further experiments were performed on DHE-containing PC. When the spectra of lipophilic extract of lipofuscin and oxidised DHE (OxDHE) were normalised to the same values at 284 nm, it appeared that the OxDHE could contribute as much as 62% and 17% to the absorption of light by lipofuscin extract above 264 nm and 390 nm, respectively (Figure 1E). It cannot be ruled out that only certain components from the mixture remain within the lipofuscin granule, while others diffuse out. 

### 2.2. Nanosecond Photoexcitation of Oxidised DHE Leads to the Generation of Similar Transient Species as That Generated by Photoexcited Lipofuscin

To determine whether oxidised DHE mixture contains products photoexcitation of which leads to the generation of an excited triplet state similar to the excited triplet state generated by RPE lipofuscin, we used nanosecond laser flash photolysis combined with kinetic absorption spectroscopy to monitor photoformation of short-lived transient products, such as excited triplet states and free radicals. This technique uses a short laser pulse to excite the chromophores in the sample which absorb the wavelength emitted by the laser [64]. As a result, the chromophore can be photolyzed meaning that the electron from the highest occupied molecular orbital (HOMO) is promoted into the lowest unoccupied molecular orbital (LUMO). Typically, in this primary process, the photoexcited electron retains its spin. In the case of molecules with photosensitising properties, the next step on the way of electron return to the ground state is the change of spin of that electron. As a result, an excited triplet state is formed, which tends to be three orders of magnitude longer-lived than the excited singlet state it originated from. Due to its longer lifetime and changed oxidation potential, the triplet state can interact with other molecules, leading to energy or electron transfer and thereby can cause photosensitised generation of reactive molecules. Photolysis of deoxygenated solution of oxidised DHE induced by 5 nm laser pulse of 355 nm wavelength led to the rapid, completed within 180 ns, photoformation of a transient species with broad difference absorption spectrum with a maximum at about 420 nm (Figure 2A). Fitting of an exponential decay curve to the decay of the transient species at 420 nm provided a fitted parameter corresponding to the lifetime of 12 µs (Figure 2B). The rate of decay of the transient was concomitant with the rate of partial recovery of the ground state absorption monitored at 340 nm (Figure 2B).

The observed transient species was similar to the transient species formed as a result of photoexcitation of lipofuscin with 355 nm laser pulse—it also exhibits an absorption maximum at about 420 nm and decays on the same time scale, concomitant with the recovery of the ground state [7,9]. 

The lipofuscin transient species was identified as an excited triplet state based on its ability to transfer its energy to carotenoid as evidenced by the recovery of the ground state with concomitant formation of the carotenoid triplet state [9]. To determine whether the transient species formed by oxidised docosahexaenoate is a triplet state, we employed also a carotenoid, zeaxanthin, which does not undergo an efficient intersystem crossing itself so does not form a detectable zeaxanthin triplet state upon direct photoexcitation [65]. Zeaxanthin triplet state energy is, as for other carotenoids, relatively low in comparison to other biologically relevant molecules; therefore, zeaxanthin can serve as an energy acceptor from excited triplet states of higher energies [65]. Photoexcitation of OxDHE in the presence of zeaxanthin led to a faster decay of the 420 nm-absorbing transient than in the absence of zeaxanthin, completed within 3 µs, and the concomitant formation of zeaxanthin triplet state with a characteristic maximum at about 510–520 nm (Figure 3), indicating that the transient species formed by photoexcited oxidised DHA is a triplet state. 

The excited triplet state of lipofuscin is rapidly quenched by oxygen with the bimolecular rate constant of quenching of 1.2 × 10^9^ M^−1^s^−1^ in hexane [9]. Therefore, we investigated the interactions of oxygen with the excited triplet state of OxDHE. In the presence of oxygen, the spectral characteristics of the transient absorption spectra were similar to those observed in the absence of oxygen, but the decay of the transient and the recovery of the ground state occurred on a much shorter time scale of 2 µs (Figure 4) than in the absence of oxygen, in which the process was not completed within 60 µs (Figure 3). The lifetime of the OxDHE triplet state in the presence of oxygen was 0.26 µs, while its reciprocal, the rate of decay, was 3.92 × 10^6^ s^−1^ (Figure 4), indicating that the bimolecular rate constant of quenching of the OxDHE triplet state by oxygen in acetone is about 1.6 × 10^9^ M^−1^s^−1^, which is close to the diffusion-controlled limit in this solvent (2.0 × 10^10^ M^−1^s^−1^; [66]) and similar to the rate of quenching by oxygen of the lipofuscin triplet state in hexane of 1.2 × 10^9^ M^−1^s^−1^ [9].

Altogether, oxidised DHE forms an excited triplet state with similar spectral properties as the excited triplet state of lipofuscin, and, similarly to the lipofuscin triplet state, it can be quenched by carotenoid and oxygen.

### 2.3. Photoexcitation of Oxidised DHE under Aerobic Conditions Leads to the Generation of Singlet Oxygen

In the case of lipofuscin, at least a part of the interaction of the excited triplet state with oxygen leads to the energy transfer to oxygen and formation of electronically excited singlet state of molecular oxygen O_2_(^1^Δ_g_) [7,9]. In comparison with oxygen in its ground state (which is a triplet state), the singlet oxygen is much more reactive and can oxidise unsaturated lipids and other biomolecules [67]. Therefore, to test whether or not the photoexcitation of OxDHE in the presence of oxygen also leads to the generation of singlet oxygen, we used a direct method of singlet oxygen detection based on monitoring the characteristic singlet oxygen phosphorescence at 1270 nm, which is emitted when O_2_(^1^Δ_g_) returns to the ground state [9]. Photoexcitation of aerated solution of OxDHE with either 405 nm or 495 nm light resulted in a broad-band emission spectrum including the characteristic for singlet oxygen emission maximum at 1270 nm (Figure 5). The emission maximum at 1270 nm disappeared upon saturation of the sample with argon, whereas the broad emission below and around the 1270 nm considerably increased, suggesting that the 1270 nm emission is due to singlet oxygen returning to the ground state, whereas the other emission corresponds to OxDHE luminescence, which increases in the absence of oxygen. Overall, the emission spectrum was similar to that observed for lipofuscin photoexcited with 488 nm light, in which the 1270 nm maximum can also be observed on a much broader luminescence background [9].

Consistently, photoexcitation of OxDHE with a 5 ns laser pulse also resulted in oxygen-dependent near-infrared emission, which decayed to the baseline within about a hundred microseconds (Figure 6A,B). In the air- or oxygen-saturated benzene, excitations of OxDHE with either 355 nm or blue-light-induced near-infrared emissions with lifetimes of 29 µs which is characteristic for singlet oxygen lifetime in this solvent [9]. The emission was reduced below the detection level upon saturation of samples with argon. Altogether, the emission maximum and the lifetime of infrared luminescence, and the dependence of the emission on the presence of oxygen, unambiguously indicate that singlet oxygen phosphorescence contributes to the infrared emission observed upon photoexcitation of OxDHE.

Next, the quantum yields of singlet oxygen generation were determined using tetraphenylporphyrin (TPP) and all-*trans*-retinal as standards with known quantum yields of singlet oxygen generation (Figure 6B). The quantum yield corresponds to the ratio of singlet molecules generated to the number of photons absorbed. The quantum yields were calculated (according to the formula included in Methods) by comparing the slopes of the initial intensities of singlet oxygen emission as a function of laser energy (Figure 6B,D).

Saturation of samples with oxygen increases oxygen concentration about fivefold, in comparison with air-saturated samples [66], which can facilitate the energy transfer from the excited triplet state to oxygen, especially if the sample contains other molecules which can serve as energy acceptors from the excited states and thereby compete with oxygen. For photoexcitation with 355 nm, the quantum yields of singlet oxygen photogenerated by OxDHE were 0.22 ± 0.01 and 0.26 ± 0.01 in the air- and oxygen-saturated solutions, respectively (Figure 6B). The quantum yields of singlet oxygen generation in samples of OxDHE photoexcited with 425 nm light were the same, independently of whether the samples were saturated with air or oxygen (Figure 6C). The effects of increasing oxygen concentrations were much more pronounced in the case of lipofuscin, in which the quantum yields of singlet oxygen generation increase upon saturation with oxygen by 88 and 80%, respectively, for excitation with 355 nm and blue light [9]. It has been previously shown that in lipofuscin granules, A2-E can act as an energy acceptor from other photoexcited chromophores [68].

For different samples of OxDHE, the quantum yields of singlet oxygen generation upon photoexcitation with 425 nm light varied from 0.11 to 0.13 (Figure 6D), indicating that similar oxidation conditions can produce mixtures of oxidation products with different ratios of products with greater and smaller photosensitising properties.

The quantum yields of singlet oxygen generation by OxDHE were 2.8- and 2.4-fold greater than the corresponding quantum yields measured for lipofuscin photoexcited with 355 nm and blue light in the same solvent, air-saturated benzene [9]. In oxygen saturated benzene, the quantum yields of singlet oxygen generated by OxDHE were still 1.7 and 1.2–1.4-fold greater than those of lipofuscin photoexcited with 355 nm or blue light, respectively. This is consistent with the results shown in Figure 1E, suggesting that a substantial contribution to absorption spectra of lipofuscin can be made by chromophores with a much smaller ability of photosensitised generation of singlet oxygen than OxDHE.

### 2.4. Photoexcitation of Oxidised DHE in the Presence of DMPO Leads to Spin-Trapping of Free Radicals Similar as Those Generated by Photoexcited Lipofuscin

Exposure to visible light of lipofuscin granules suspended in air-saturated phosphate buffer in the presence of a spin trap, 5,5-dimethyl-1-pyrroline N-oxide (DMPO) leads to the formation of DMPO-^●^OOH and DMPO-^●^OH adducts, accumulation of which could be diminished by superoxide dismutase (SOD) and catalase, suggesting that they are formed due to interaction of DMPO with the superoxide and hydroxyl radicals, respectively [6]. To determine whether similar adducts can be formed as a result of photoexcitation of OxDHE, it was suspended as liposomes in phosphate buffer and exposed to visible light. During the exposure, the characteristic spectra of DMPO-^●^OH adducts appeared (Figure 7). The accumulation of these adducts was substantially inhibited in the presence of superoxide dismutase and catalase. Superoxide dismutase catalyses the dismutation of superoxide radicals into hydrogen peroxide [69]. It is also well established that the hydrogen peroxide undergoes decomposition in the presence of redox-active metal ions generating the hydroxyl radical (known as Fenton reaction), and that the DMPO-^●^OOH adduct spontaneously undergoes decomposition to DMPO-^●^OH adduct [69]. The addition of inactivated catalase or bovine serum albumin exerted much smaller effects than the active enzymes, indicating that the DMPO-^●^OH signals were, in part, due to decomposition of DMPO-^●^OOH, and in part, due to the interaction of DMPO with hydroxyl radical formed as a result of decomposition of hydrogen peroxide. Again, these results are very similar to those recorded for lipofuscin.

The interactions of superoxide with DMPO, and the formation of more stable adducts than in an aqueous solvent, can be facilitated by using as the solvent the dimethylsulphoxide (DMSO) [13,15]. Indeed, the exposure to visible light of OxDHE liposomal extract solubilised in air-saturated DMSO led to the accumulation of DMPO-^●^OOH adducts (Figure 8A). Irradiation of samples containing DHE also resulted in accumulation of DMPO-^●^OOH adducts but with the rate substantially smaller than that for OxDHE (Figure 8A,B). Irradiation of samples in the absence of oxygen prevented the accumulation of DMPO-^●^OOH adducts and led to the formation of DMPO adducts with carbon-centred radicals, DMPO-^●^CR (Figure 8A).

To compare the quantum yields of generation of superoxide by OxDHE and lipofuscin, we compared the kinetics of DMPO-^●^OOH adducts accumulation mediated by irradiated OxDHE and riboflavin with an optically matched solution at the irradiation wavelength of 404 nm (Figure 8C). The initial rate of increase in DMPO-^●^OOH signal photogenerated by OxDHE was 2.8-fold smaller than that photogenerated by riboflavin. Rate of growth of DMPO-^●^OOH signal photogenerated under similar conditions by lipophilic extract of lipofuscin has been shown to be 10.2 times smaller than that for riboflavin [14], indicating that the quantum yield of generation of superoxide by OxDHE is 3.6-fold greater than for lipofuscin extract. Taking that the quantum yield of photogeneration of superoxide by riboflavin is 0.009 [70], it allows calculating the quantum yields of superoxide generation by OxDHE and lipofuscin extract as 0.0035 and 0.00096, respectively. Again, these results are consistent with results on the contribution of OxDHE to the absorption spectrum of lipofuscin and suggest that OxDHE in lipofuscin may contribute to its photoreactivity while some other chromophores contribute to the absorption of blue light but exhibit lower photoreactivity than OxDHE.

### 2.5. Photoexcitation of Oxidised DHE Leads to Oxygen Consumption with a Similar Irradiation Wavelength-Dependence as That for Lipofuscin

Exposure of lipofuscin to light leads to the irradiation wavelength-dependent consumption of oxygen with its rates monotonically increasing towards shorter irradiation wavelengths [6,12,15]. To investigate the susceptibility of OxDHE liposomes to oxidation and photooxidation, we monitored oxygen concentrations in liposomal suspensions in dark and during exposure to narrow-band light (Figure 9A). In the dark, liposomes made either from OxDHE or freshly opened DHE exhibited a detectable oxygen consumption, with oxygen depletion occurring faster in the suspension of OxDHE than in DHE liposomes. During irradiation with ultraviolet or visible light, the oxygen consumption was considerably accelerated in both types of liposomes.

To compare the susceptibility of DHE and OxDHE liposomes to photooxidation as a function of irradiation wavelength, the initial rates of oxygen uptake were normalised to equal fluxes of incident photons (Figure 9B). The resulting action spectrum reflects the absorption spectrum of chromophores responsible for light-induced oxidation. The action spectrum for DHE shows that once exposed to the air, DHE can undergo rapid oxidation. In the case of both action spectra, the efficiencies of photooxidation increase with decreasing irradiation wavelength. The longest irradiation wavelengths for which the rates of oxygen consumption were greater than the rates for liposomes incubated in the dark were 406 and 480 nm, respectively, for DHE and OxDHE. The action spectrum of photooxidation mediated by OxDHE is similar to the action spectrum of photooxidation mediated by lipofuscin or lipophilic extract of lipofuscin [6,12,15].

## 3. Materials and Methods

### 3.1. Reagents

Chemicals, at least analytical grade, including docosahexaenoic acid (DHA), 5,5-dimethyl-1-pyrroline N-oxide (DMPO), and dimethyl sulfoxide (DMSO), were purchased from Sigma (Sigma, St. Louis, MO, USA) or Fisher Scientific (Loughborough, UK) unless stated otherwise. Phospholipids containing DHE, namely, 1,2-di-(4Z,7Z,10Z,13Z,16Z,19Z-docosahexaenoyl)-sn-glycero-3-phosphocholine (Di22:6PC) and 1-hexadecanoyl-2-(4Z,7Z,10Z,13Z,16Z,19Z-docosahexaenoyl)-sn-glycero-3- (16:0-22:6PC) were from Avanti Polar Lipids (Alabaster, AL, USA). Analytical grade zeaxanthin (>99% purity was a generous gift from DSM (Basel, Switzerland). The spin probe 4-protio-3-carbamoyl-2,2,5,5-tetraperdeuteromethyl-3-pyrroline-1-yloxy (mHCTPO) was a gift from Professor Howard J. Halpern (University of Chicago, Chicago, IL, USA) and was used as received. Chromatography grade acetone, acetonitrile, benzene, chloroform hexane, and methanol were from Fisher, Merck, or VWR International, and used as supplied.

### 3.2. Isolation and Purification of RPE Lipofuscin

Research on human tissue was approved by the School Research Ethics Audit Committee, School of Optometry and Vision Sciences, Cardiff University (SREAC Project Number 1406 approved on 18/02/2016). Lipofuscin was isolated and purified from human eyes 41 to 80 years old, as described previously [15]. In short, RPE cells in an eye-cup with removed anterior segment and vitreous were gently brushed into PBS, aspirated into Eppendorf tubes, and stored frozen at −80 °C until sufficient numbers of RPE cells were collected. Lipofuscin granules were isolated by differential centrifugation and purified by centrifugation on sucrose gradient at 103,000× *g*.

### 3.3. Preparation of Liposomes

Multilamellar lipid vesicles (liposomes) were prepared, as described previously [15,71], under anaerobic conditions and dim red light. In short, Di22:6PC or 16:0-22:6PC was dissolved in argon-saturated chloroform, followed by quick evaporation of the solvent using a rotary evaporator. To remove the remnants of chloroform, the lipid film was dried under vacuum for at least an hour and then exposed to the air at 37 °C for 28 days, followed by hydration in phosphate buffer saline (PBS), pH 7.1 at 37 °C. DHE liposomes, which were meant to be kept not oxidised, were hydrated immediately after the formation and drying of the lipid film with argon-saturated PBS.

### 3.4. Autooxidation of DHA and DHE and Its Monitoring by Spectrophotometry

Immediately after opening the vial with DHA, an aliquot was taken out and used immediately for measurement of optical absorption spectra without the addition of any solvents. Another aliquot was weighted and solubilised in acetonitrile:methanol (1:1) at a concentration of 100 mg/mL and absorption spectrum was also measured. The remaining DHA was aliquoted into Eppendorf tubes (about 100 mg/tube), weighted, and exposed to the air at 37 °C to enable autooxidation. DHA depletion and formation of oxidation products were monitored by measurements of optical absorption using U-2800 UV–VIS spectrophotometer (Hitachi) after selected times since opening the DHA vial and solubilisation of DHA/oxidised DHA in acetonitrile:methanol (1:1).

To solubilise lipids from 16:0;22:6PC liposomes, the liposomes were extracted in chloroform/methanol mixture [15], followed by collecting the chloroform-enriched fraction, taking an aliquot for absorption measurements, and drying the rest under a stream of argon so to be resolubilised in acetone, benzene, or DMSO. All procedures were performed under dim red light.

The hydration of oxidised Di22:6PC lipid film resulted in its good solubilisation, and hence the absorption spectra were measured in PBS.

### 3.5. Time-Resolved Detection of Transient Species Formed by Nanosecond Laser Flash Photolysis of Oxidised DHE

Nanosecond laser flash photolysis, combined with kinetic absorption spectroscopy, was used to monitor transient products formed in the mixture of oxidation products of DHE or 16:0;22:6PC extracted from liposomes and solubilised in acetone [9,72]. The photolysis was induced either by a 5-ns laser pulse of 355 nm wavelength from Continuum Surelite II-10 Q-switched Nd:YAG laser (Photonic Solutions Plc., Edinburgh, UK) or by selected wavelengths from a range of 410–2550 nm obtained from the tuneable Continuum Panther type-II optical parametric oscillator pumped by the third harmonic of Nd:YAG laser (Photonic Solutions Plc., Edinburgh, UK), as described previously [73]. Absorbance changes were monitored by LKS.60 nanosecond time-resolved spectrometer (Applied Photophysics, Leatherhead, UK). In short, a 150 W xenon arc lamp (OSRAM X/150W/CR/OFR, MGC Lamps Ltd., Suffolk, UK), equipped with a booster, was used as a source of monitoring light. Two monochromators—one in between the sample and the lamp source, and the second in between the sample and the photomultiplier—provided a monochromatic monitoring beam to diminish sample photodegradation and minimise scattered light reaching the photomultiplier. The signal from the photomultiplier was coupled to the Agilent 54830B digitiser (Agilent Technologies Ltd., Stockport, UK) interfaced with the RISC workstation (Applied Photophysics, Leatherhead, UK). Zeaxanthin was used as a triplet energy acceptor [65].

### 3.6. Detection of Singlet Oxygen Phosphorescence

The singlet oxygen phosphorescence spectra were measured using a steady-state spectrophotometer, as described previously [74]. In short, the samples containing OxDHE were excited with light from 450 W mercury lamp and an interference filter transmitting light at 405 nm or 495 nm. The luminescence was monitored at a 90° angle to the excitation beam, using a monochromator and a germanium diode cooled with liquid nitrogen.

The time-resolved detection of formation and decay of the characteristic singlet oxygen phosphorescence at 1270 nm following nanosecond laser flash photolysis was performed, as described previously [9,72,73]. All-*trans*-retinal and tetraphenylporphyrin (TPP) were used as standards with known quantum yields of singlet oxygen generation of 0.30 +/− 0.04 and 0.63, respectively [9,75]. To determine the quantum yields, solutions of oxidised 16:0-22:6PC and all-*trans*-retinal, with the same absorbances at the excitation wavelength, were subjected to the laser pulse excitation. The initial emission intensities, extrapolated from the experimental points fitted to exponential decay curves, were measured as a function of laser energy. The slopes of the initial linear portion of the curves were used for calculation of singlet oxygen quantum yield generated by the oxidised 16:0;22:6PC according to the following formula:Φ_OxDHE_ = Φ_Ral_ (a_OxDHE_/a_Ral_),
where Φ_OxDHE_ denotes the quantum yield of singlet oxygen generation by the OxDHE, Φ_Ral_ is the quantum yield of singlet oxygen generation for all-*trans*-retinal, a_OxDHE_ is the fitted slope for OxDHE; and a_Ral_ is the fitted slope for all-*trans*-retinal.

### 3.7. Continuous Irradiation with Narrow- or Broad-Band Light

Suspensions of liposomes were irradiated either with broadband visible light derived from a compact-arc high-pressure xenon lamp (PhotoMax 150 W; Oriel, Darmstadt, Germany) with a combination of cut-off (<390 nm) and copper sulphate filters (effective spectral range: 390–620 nm, fluence rate ~ 27 mW/cm^2^), or with a narrow-band light using a combination of broadband, cut-off and interference filters [6,12,14,15]. Fluence rates of broad-band irradiation were measured using a YSI radiometer, model 65A (Yellow Spring Instruments Co., Yellow Springs, OH, USA), and of narrow-band irradiation using a calibrated silicon photodiode (Hamamatsu, Photonics, K.K., Hamamatsu City, Japan) inside the resonant cavity of electron spin resonance (ESR) spectrometer.

### 3.8. Electron Spin Resonance (ESR) Spin Trapping

ESR spin trapping was used to monitor photogeneration of short-lived free radicals during irradiation of samples with visible light in a flat ESR cell (Wilmad Glass Co., Vineland, NJ, USA) with an optical pathlength of 0.3 mm in situ in the resonant cavity, as described previously [6,13,14,15]. Samples contained suspensions of oxidised 16:0;22:6PC liposomes in phosphate-buffered saline (PBS) or liposomal extracts of oxidised DHA solubilised in DMSO in the presence of 100 mM DMPO used as a spin trap. ESR spectra were recorded using the ESR spectrometer (ESP 300E; Bruker, Billerica, MA, USA) operating at 9.5 GHz with 100-kHz field modulation. To distinguish between primary and secondary products, the time course of formation of DMPO adducts and their decay was followed both in the dark and during irradiation with light.

### 3.9. Light-Dependent Oxygen Consumption

Kinetics of oxygen concentration changes in irradiated samples were measured by ESR oximetry, as described before [6,13,14,15,76]. In brief, the liposomal suspension in PBS, with an addition of 0.1 mM mHCTPO used as the nitroxide spin probe, was placed in a flat quartz cell (optical pathlength of 0.3 mm) in a resonant cavity, and ESR spectra of mHCTPO were collected during their illumination in situ, at ambient temperature. A calibration curve was used (based on measuring mHCTPO spectra of PBS saturated with various mixtures of oxygen, in which oxygen concentration was determined by oxygen electrode) to calculate oxygen concentration based on spectral characteristics of mHCTPO. The initial rates of oxygen uptake were obtained from the slopes of the linear fits to the data points corresponding to the initial oxygen depletion, in which the oxygen concentration decreased linearly with time.

### 3.10. Irradiation Wavelength-Dependence of Photooxidation

To obtain the action spectra of photo-induced oxygen consumption, the initial rates of oxygen uptake were normalised to equal numbers of incident photons, as described previously, that is, the rate of oxygen consumption, expressed in mM/s, was divided by the irradiation wavelength (in nm) and by the fluence rate (mW/cm^2^) [6,13,14,15].

## 4. Conclusions

Our results demonstrate that autooxidation of docosahexaenoate results in the formation of products with the absorption spectrum extending into the UVA and visible light, with absorption monotonically decreasing with increasing wavelength, similarly to the absorption characteristics of lipofuscin granules and lipophilic extract of lipofuscin [6,12,15].

Photoexcitation of the mixture of these products leads to the intersystem crossing and formation of an excited triplet state with similar spectral characteristics to those of the excited triplet state of lipofuscin. Similar to the lipofuscin triplet state, the triplet state formed as a result of photoexcitation of oxidised DHE is quenched by oxygen with the bimolecular rate constant close to the diffusion-controlled limit. Similar to lipofuscin, as a result of the interaction of the excited triplet state of OxDHE with oxygen, an energy transfer occurs to molecular oxygen leading to the photoformation of the electronically excited state of molecular oxygen, singlet oxygen ^1^O_2_(^1^Δ_g_). Similar to lipofuscin, photoexcitation of oxidised DHE under aerobic conditions leads to the generation of superoxide radicals.

The quantum yields of singlet oxygen and superoxide photogenerated by OxDHE are greater than those of lipofuscin. However, it has been established that lipofuscin includes chromophores, such as A2-E, which photosensitising activity is much smaller than that of lipofuscin or OxDHE, but they do contribute to the absorption of light. Assuming 17% contribution of OxDHE to the absorption of visible light by lipophilic extract of lipofuscin, and all the remaining 83% absorption allocated to A2-E, the quantum yields of singlet oxygen generation of 0.12 and 0.004 for OxDHE and A2-E, respectively, gives 6.14 times greater contribution to singlet oxygen generation by OxDHE than by A2-E.

Importantly, the action spectra of photooxidation mediated by lipofuscin and OxDHE show similar wavelength-dependences when normalised to the equal number of incident photons, indicating that OxDHE can be an important contributor to lipofuscin photoreactivity.

The accumulation of RPE lipofuscin is mostly a result of incomplete lysosomal digestion of photoreceptor outer segments, which contain high concentrations of DHE, and therefore, DHE is abundant in RPE lipofuscin. However, it cannot be excluded that products of DHE oxidation can contribute to photoreactivity of lipofuscin in other cell types exposed to light, such as retinal neurons in the retina or keratinocytes in the skin [77,78].

In conclusion, our results demonstrate that products of DHE oxidation exhibit photosensitising properties which can account for the photoreactivity of retinal lipofuscin. These results are important because they point to a need for investigation into molecular components of lipofuscin because none of the structures of DHE oxidation products reported so far appear to have the ability to absorb visible light. Identification of the molecular species responsible for photosensitising properties of oxidised DHE may help to develop an adequate pharmacological intervention to counteract its formation and/or photoreactivity.

## Figures and Tables

**Figure 1 ijms-22-03525-f001:**
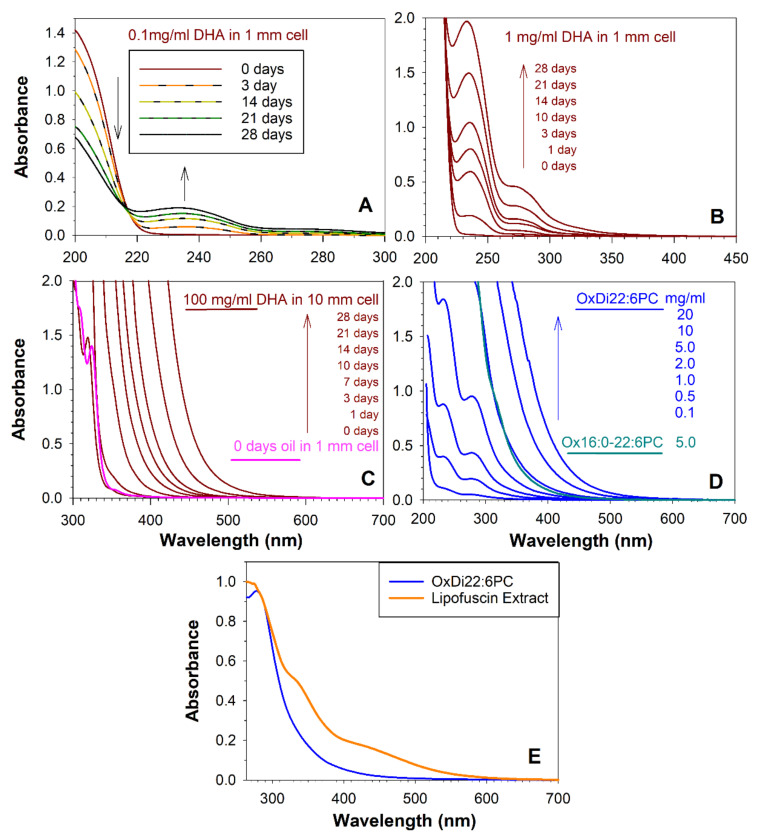
Autooxidation of docosahexaenoic acid (DHA) and docosahexaenoate (DHE) in phospholipids results in the formation of products absorbing ultraviolet and visible light. Absorption spectra of DHA after indicated times of autooxidation of DHA as an oil under aerobic conditions at 37 °C, then solubilised in acetonitrile:methanol (1:1), and monitored at a concentration of 0.1 mg/mL DHA in a cell with 1 mm optical pathlength (**A**), 1 mg/mL in a cell with 1 mm optical pathlength (**B**), and 100 mg/mL in a cell with 10 mm optical pathlength (**C**). Graph C also includes the absorption spectrum of pure DHA oil recorded in a cell with 1 mm optical pathlength. (**D**) Absorption spectra of autooxidised for 28 days at 37 °C phosphatidylcholines with one (Ox16:0-22:6PC) or two (OxDi22:6PC) DHE acyl chains at indicated concentrations of the initial phospholipid recorded in a cell with 10 mm optical pathlength. Ox16:0-22:6PC was solubilised in chloroform, while OxDi22:6PC was solubilised in phosphate buffer saline (PBS). (**E**) Comparison of the absorption spectra of OxDi22:6PC with lipophilic extract of lipofuscin solubilised in chloroform normalised to the same value at 284 nm.

**Figure 2 ijms-22-03525-f002:**
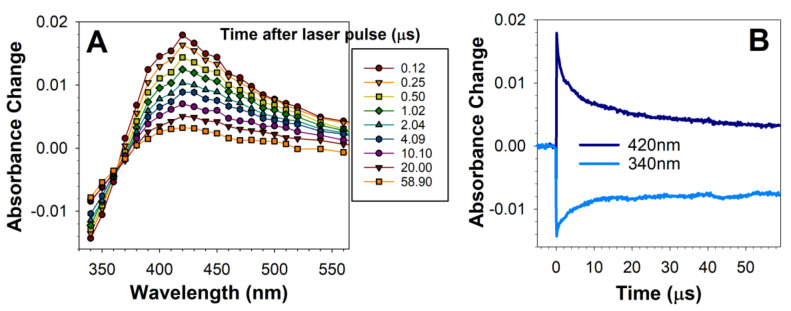
Photoexcitation of oxidised DHE with a 5 ns laser pulse results in the formation of a transient species with an absorption maximum at 420 nm. Representative absorption spectra of transient species generated at indicated times after photoexcitation of oxidised DHE (**A**) and kinetics of transient species formation and decay recorded at 420 nm, and sample bleaching and recovery monitored at 340 nm (**B**). Oxidised DHE was solubilised in argon-saturated acetone and exposed, at time 0, to a 5 ns laser pulse at 355 nm wavelength. The transient species exhibits an absorption maximum at about 420 nm and decays concomitantly with the partial recovery of absorption at 340 nm.

**Figure 3 ijms-22-03525-f003:**
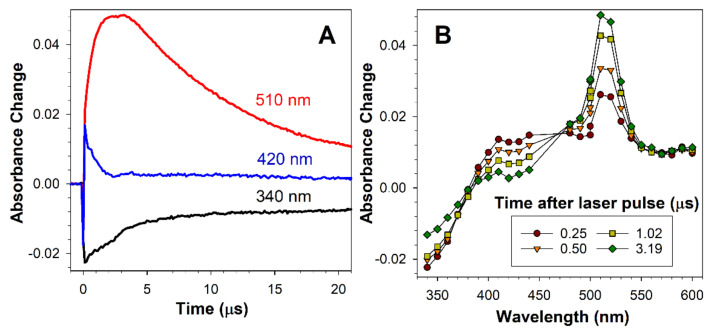
The transient species formed after photoexcitation of DHE transfers is an excited triplet state. Representative kinetics (**A**) and absorption spectra of a transient species generated at indicated times after photoexcitation of oxidised DHE with a 5 ns laser pulse of 355 nm wavelength in argon-saturated acetone in the presence of zeaxanthin (**B**). The lifetime of 420-nm-absorbing species is shortened in the presence of zeaxanthin, in comparison with its lifetime in the absence of zeaxanthin, shown in Figure 2B, and decays with concomitant formation of zeaxanthin triplet state with a maximum at 510–520 nm. Zeaxanthin contributes to the absorption of light at 340 nm, which results in more complex kinetics of the recovery of the ground-states than in the absence of zeaxanthin.

**Figure 4 ijms-22-03525-f004:**
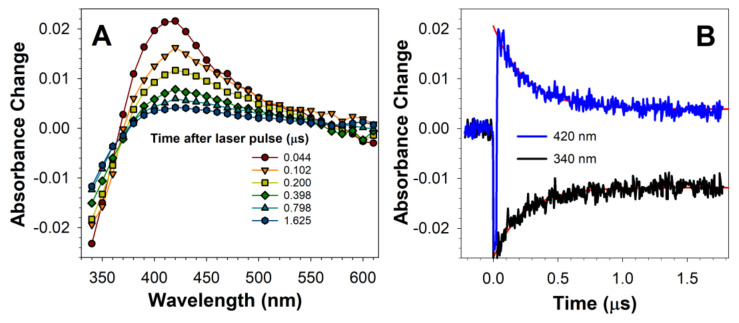
The transient species formed after photoexcitation of DHE is quenched by oxygen. Representative absorption spectra of transient species generated at indicated times after photoexcitation of oxidised DHE (**A**), and kinetics of transient formation and decay recorded at 420 nm, and sample bleaching and recovery monitored at 340 nm (**B**). Oxidised DHE was solubilised in air-saturated acetone and exposed a 5 ns laser pulse of 355 nm wavelength. The transient species exhibits an absorption maximum at about 420 nm and decays concomitantly with partial recovery of absorption at 340 nm but on a much faster time scale than in the de-oxygenated solution, as shown in Figure 2.

**Figure 5 ijms-22-03525-f005:**
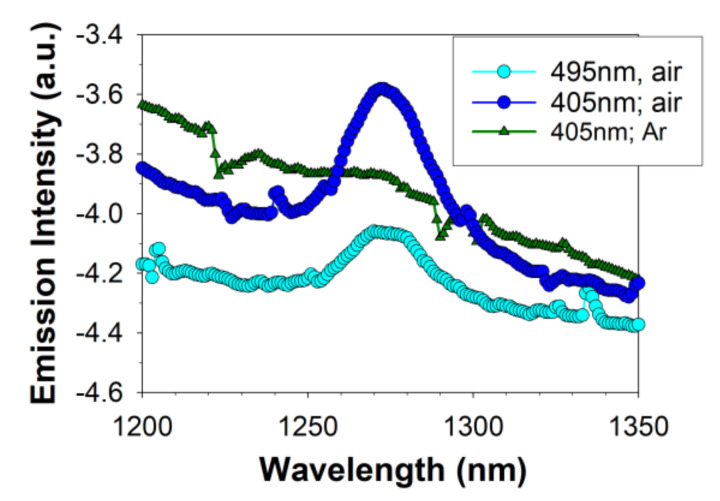
Photoexcitation of oxidised DHE (OxDHE) with blue light results in oxygen-dependent infrared emission with a maximum at 1270 nm. Near-infrared emission spectra from OxDHE solutions, saturated with air or argon (Ar), were collected during exposure to either 405-nm or 495-nm light.

**Figure 6 ijms-22-03525-f006:**
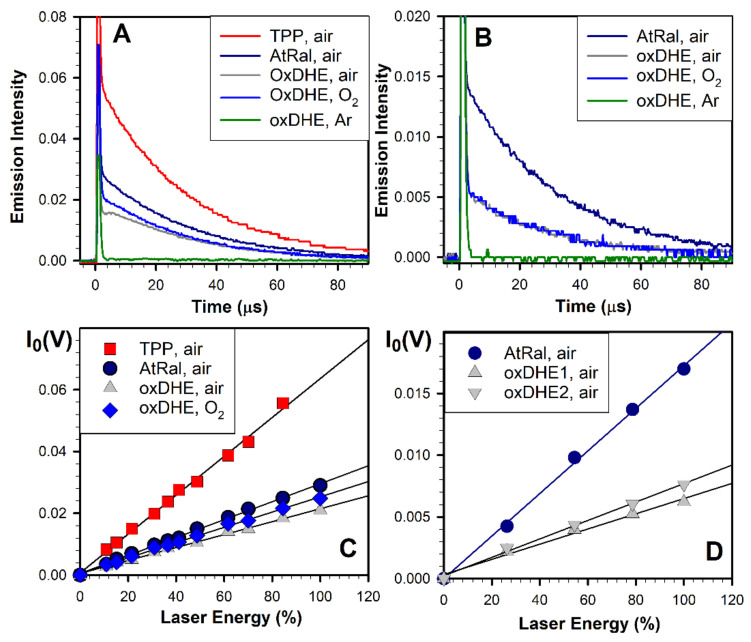
Photoexcitation of oxidised DHE (OxDHE) with a 5 ns laser pulse results in oxygen-dependent infrared emission at 1270 nm decaying with the characteristic for singlet oxygen lifetime and relatively high quantum yields. Representative kinetics of formation and decay of near-infrared emission after photoexcitation of OxDHE with a 355 nm (**A**) and 425 nm (**B**) 5 ns laser pulse. OxDHE was solubilised in benzene saturated with either air, oxygen, or argon. All-*trans*-retinal (AtRal) and tetraphenylporphyrin (TPP) were used as standards of known quantum yields of singlet oxygen generation in benzene. To ensure the same number of absorbed photons by samples containing OxDHE and standards, the samples were diluted to have their absorbances adjusted to the same value at the excitation wavelength. The decays of singlet oxygen phosphorescence were fitted with exponential decay functions to obtain, as fitted parameters, the initial intensities of singlet oxygen phosphorescence. (**C**,**D**) The initial intensities, I_0_ plotted as a function of laser energy. Straight lines were fitted to datapoints within the linear range of I_0_ increase with laser energy. The ratio of the quantum yields of singlet oxygen generation by OxDHE and a standard is the same as the ratio of their corresponding slopes of the fitted straight lines, which allows the calculation of the quantum yield of singlet oxygen generation by OxDHE.

**Figure 7 ijms-22-03525-f007:**
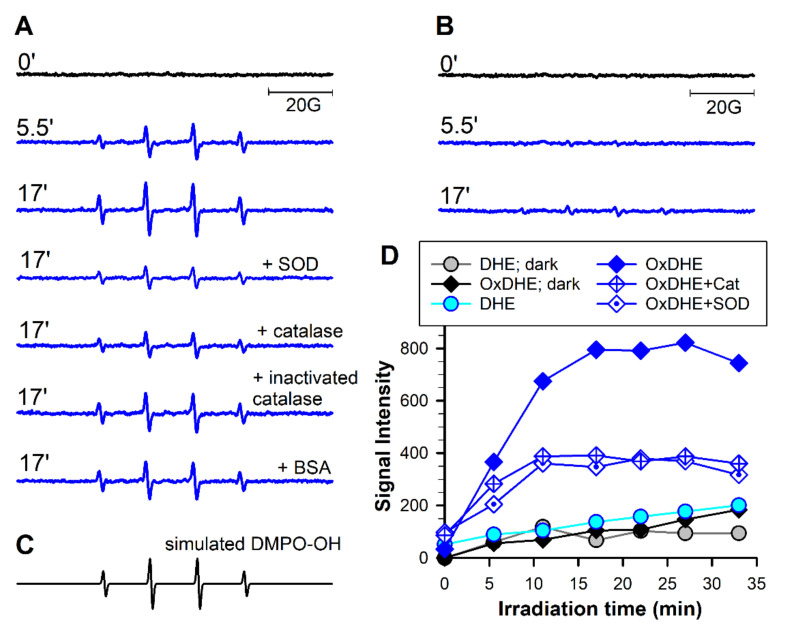
Photoexcitation of OxDHE liposomes with visible light in the presence of 5,5-dimethyl-1-pyrroline N-oxide (DMPO) as the spin trap DMPO results in superoxide dismutase (SOD)- and catalase-dependent accumulation of DMPO-^●^OH adducts. (**A**,**B**): Representative electron spin resonance (ESR) spectra acquired from samples of liposomal suspensions made from oxidised (OxDHE, **A**) or freshly opened 16:0-22:6PC (DHE, **B**) exposed in the presence of 0.1 DMPO to 27 mW/cm^2^ visible light inside the resonant cavity of ESR spectrometer. The spectra were acquired at indicated irradiation times in the absence and presence of 0.1 mg/mL superoxide dismutase (SOD), catalase, catalase inactivated by irradiation with UV light, or bovine serum albumin (BSA). (**C**) a simulated spectrum of DMPO adduct with the hydroxyl radical (DMPO-^●^OH) allowing for identification of the adduct in the experimental spectra. (**D**) representative kinetics of DMPO-^●^OH adduct accumulation in the suspension of liposomes made from DHE or OxDHE in the dark (as indicated in the legend) or during irradiation with light (blue and cyan symbols and lines) in the absence and presence of SOD or catalase (Cat).

**Figure 8 ijms-22-03525-f008:**
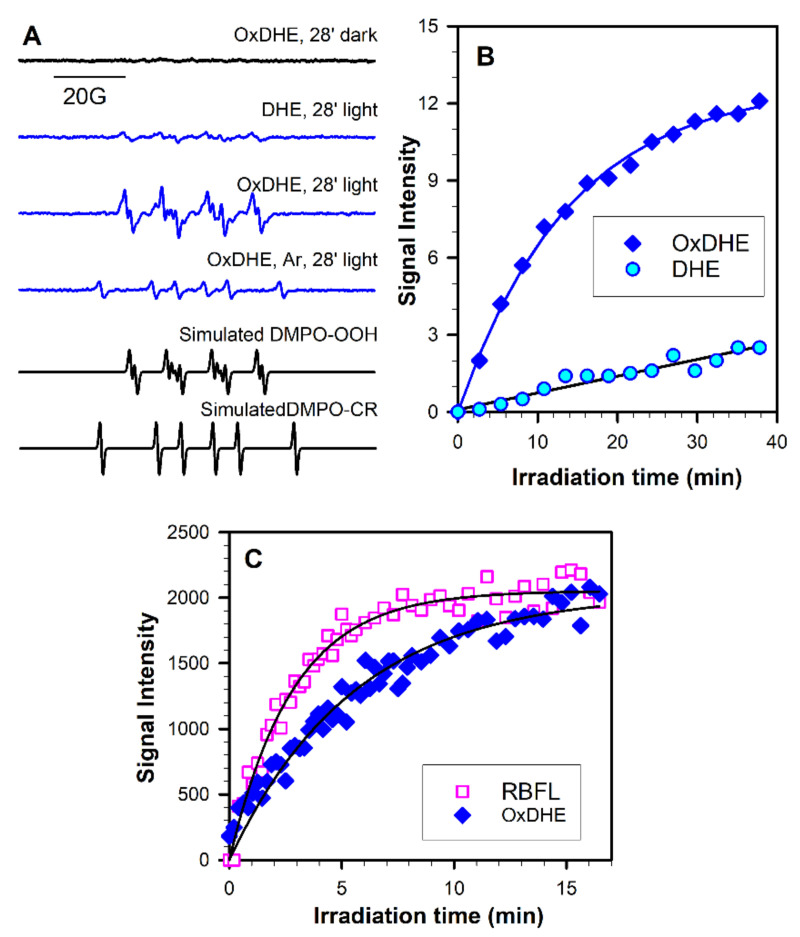
Photoexcitation of oxidised DHE solubilised in dimethylsulphoxide (DMSO) with visible light in the presence of DMPO as the spin trap results in oxygen-dependent accumulation of DMPO-^●^OOH adducts with a quantum yield 2.8-fold smaller than for riboflavin. (**A**) Representative ESR spectra acquired from samples of lipophilic extracts from liposomes made from oxidised (OxDHE) or freshly opened 16:0-22:6PC (DHE) solubilised in DMSO:benzene (9:1) and exposed to 27 mW/cm^2^ visible light in the presence of 0.1 M DMPO inside the resonant cavity of ESR spectrometer. The spectra were acquired after 28 min of light exposure or incubation in the dark. The samples were saturated with air, except for the sample saturated with argon (Ar). Simulated spectra of DMPO-^●^OOH and DMPO-^●^CH adducts were created to identify the adducts in the experimental spectra as DMPO adducts with superoxide radical (DMPO-^●^OOH) and DMPO adducts with carbon-centred radical (DMPO-^●^CR). (**B**) Representative kinetics of DMPO-^●^OOH accumulation during exposure of OxDHE and DHE to visible light. (**C**) Representative kinetics of DMPO-^●^OOH accumulation during irradiation with narrow-band 404 nm light (3 mW/cm^2^) of 0.35 mM riboflavin (RBFL) or OxDHE having the same absorbances at the excitation wavelength. The initial rates of accumulation obtained from parameters of the fitted lines were used for calculation of the quantum yield.

**Figure 9 ijms-22-03525-f009:**
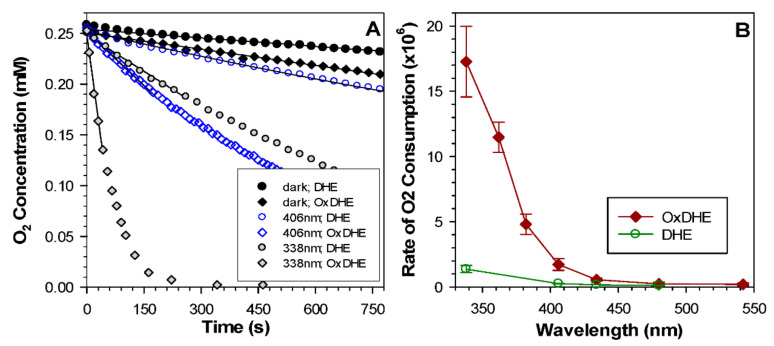
The susceptibility of oxidised DHE to photo-induced oxidation increases with decreasing irradiation wavelength. (**A**) Representative kinetics of oxygen consumption in the dark and during irradiation with narrow-band UV or blue light of indicated wavelengths in the suspension of liposomes made from oxidised (OxDHE) or freshly opened 16:0-22:6PC (DHE) in the presence of 0.1 mM 4-protio-3-carbamoyl-2,2,5,5-tetraperdeuteromethyl-3-pyrroline-1-yloxy (mHCTPO) used as a spin probe. (**B**) Irradiation wavelength-dependence of the initial rates of photo-induced oxygen uptake in the suspension of OxDHA and DHE liposomes normalised to the equal number of incident photons (the action spectra).

## Data Availability

The data presented in the manuscript are available in numerical format upon request.

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
