# Peer review of "Products of Docosahexaenoate Oxidation as Contributors to Photosensitising Properties of Retinal Lipofuscin"

_ijms, 2021, doi:10.3390/ijms22073525_

Round 1

Reviewer 1 Report

THE PAPER IT IS INTERESTING BUT IN MY OPINION THERE IS NO DESCRIPTION OF SINGLE EXPERIMENT FOR EACH ASPECT OF DHA PHARMACOCINETIC. BETTER EXPLANATION ON HOW THE EXPERIMENTS WERE PERFORMED MAY HELP COMPREHENSION BY THE READER.

TOO MANY GRAPHS ARE INCLUDED IN THE PAPER, SOME OF THEM SEEMS TO ME TO BE REPETITIVE.

Author Response

Response to Reviewer 1:

THE PAPER IT IS INTERESTING BUT IN MY OPINION THERE IS NO DESCRIPTION OF SINGLE EXPERIMENT FOR EACH ASPECT OF DHA PHARMACOCINETIC. BETTER EXPLANATION ON HOW THE EXPERIMENTS WERE PERFORMED MAY HELP COMPREHENSION BY THE READER.

We have expanded the descriptions in the amended manuscript, amended, where needed the Methods, and, where applicable, explained what the experiments were about in more detail in the Results section. The experiments were meant to re-create as closely as possible the experiments done on extracts from lipofuscin isolated from human retinal pigment epithelium, and the references to these papers were provided both in the Methods and Results.

Places in the manuscript where the changes are included (highlighted im yellow in the text):

  • The 2nd part of the 2nd paragraph in section 2.1
  • Extended 1st paragraph in section 2.2
  • Extended descriptions in paragraphs 3, 4, 5, 6 and 7 in section 2.3, extended Figure 6 caption
  • Extended descriptions in paragraphs 1, 2, and in section 2.4, extended Figure 7 and 8 captions
  • Extended 2nd paragraph in section 2.5; extended description in section 4.9

TOO MANY GRAPHS ARE INCLUDED IN THE PAPER, SOME OF THEM SEEMS TO ME TO BE REPETITIVE.

The graphs included in the manuscript illustrate and support experimental results which are described and discussed in the text. They are also needed for the direct comparisons with corresponding graphs published for lipofuscin extract. Therefore, we strongly believe that all figures included in the manuscript are essential to support the conclusions.

Reviewer 2 Report

The authors report that the lipofuscin of retina is composed of products of DHE oxidation. DHE (dihydroethidium) is a fluorogenic probe that when oxidized to ethidium (Etd+) can be excited at 535 nm.  The omega-3 fatty acid docosahexaenoic acid is known as DHA (as is docosahexaenoate). The authors describe the autooxidation of DHA in the absence of a photosensitizer or transition metal. What is responsible for initiating autooxidation of DHA? What does ‘exposure to air’ mean for the autooxidation?  The use of UV wavelengths in the study of components of retina is not physiological since wavelengths below 400 nm do not reach the retina. The authors should better account for molecules in lipofuscin.  A reading of previous literature revealed to this reviewer that portions of the work reported here are already known to the field.  Secondary oxidation products of DHA include fragments carrying aldehydes and dialdehydes that originate from the chain reaction / cleavage of fatty acid peroxides. These reactive fragments can go on to react with amine groups of the phospholipids. These adducts are typically detected at ~335 nm absorbance. Since the structures of many of the oxidation products are known, the authors should have monitored by mass. This is even more important if these are newly determined structures. In terms of structures it is known that when DHA is oxidized nonenzymatically, protective neuroprostanes are produced.  Figure 2 is not completely visible.

Author Response

Response to Reviewer 2:

The authors report that the lipofuscin of retina is composed of products of DHE oxidation.

We report on photochemical properties of unknown oxidation products of DHE which match photochemical properties of RPE lipofuscin.

DHE (dihydroethidium) is a fluorogenic probe that when oxidized to ethidium (Etd+) can be excited at 535 nm.  The omega-3 fatty acid docosahexaenoic acid is known as DHA (as is docosahexaenoate).

We defined all abbreviations in the text. We used both, docosahexanoic acid, which we abbreviated DHA, and phosphatidylcholine containing docosahexaenoate as its acyl chain(s), which we abbreviated as DHE. The spectral properties of oxidized DHE and DHA are different, with oxidized DHE resembling more closely the lipophilic extract of lipofuscin. There is no use of dihydroethidium in the manuscript so there is no reason for it to be confused with docosahexaenoate. The abbreviation of DHA is used in literature not only for docosahexaenoic acid or docosahexaenoate but also for the oxidized form of vitamin C, dehydroascorbate.

The authors describe the autooxidation of DHA in the absence of a photosensitizer or transition metal. What is responsible for initiating autooxidation of DHA? What does ‘exposure to air’ mean for the autooxidation?  

We presume that the autooxidation is initiated by traces of transition metal ions, such as iron or copper. Exposure to the air provides oxygen needed for formation of lipid peroxyl radicals, which can propagate the chain of lipid peroxidation.

The use of UV wavelengths in the study of components of retina is not physiological since wavelengths below 400 nm do not reach the retina.

We agree. In majority of normal adult human eyes, only light above 390 nm reaches the retina. For experiments where the spectral characteristics of DHA/DHE were compared with lipofuscin extract, we used both UV and visible wavelengths. The results reporting effects induced by visible light: generation of singlet oxygen and free radicals, and photooxidation, are the most physiologically relevant. The photoreactivity induced by UV light would be relevant only to people without crystalline lenses or having lenses, which have not been sufficiently exposed to UV light to oxidize the lenticular tryptophan, thereby forming there the UV-absorbing chromophores.

The authors should better account for molecules in lipofuscin.  A reading of previous literature revealed to this reviewer that portions of the work reported here are already known to the field.  Secondary oxidation products of DHA include fragments carrying aldehydes and dialdehydes that originate from the chain reaction / cleavage of fatty acid peroxides. These reactive fragments can go on to react with amine groups of the phospholipids. These adducts are typically detected at ~335 nm absorbance. Since the structures of many of the oxidation products are known, the authors should have monitored by mass. This is even more important if these are newly determined structures. In terms of structures it is known that when DHA is oxidized nonenzymatically, protective neuroprostanes are produced. 

We have expanded the description of what is known about molecular constituents of lipofuscin derived from DHA/DHE, including carboxyethyl pyrrole and its precursors in the 5th paragraph of the Introduction (highlighted in yellow). None of the DHA/DHE oxidation products identified so far absorbs visible light. We have performed extensive literature searches on PubMed and Web of Science and have not found any reports which overlap with data presented in our manuscript. In collaboration with Professor John D. Simon, Department of Chemistry, Duke University, we have used LC/MS/MS attempting in vain to separate and identify the DHE oxidation products, which do absorb visible light. We used all published methods reported in literature for separation of oxidized lipids. The longest wavelength maximum of some separated oxidation products was at about 340 nm, and we did not observe any products absorbing wavelengths corresponding to visible light.

Figure 2 is not completely visible.

We have added colour to the data points to make them more distinguishable in Fig. 2 and added colour to other figures as well.

Round 2

Reviewer 1 Report

The Authors have addressed many concerns, in my opinion too many graphs may be confusing for the reader, but if the AA ne ed all of them, the paper is sufficiently well done.

Author Response

Response to Reviewer 1

The Authors have addressed many concerns, in my opinion too many graphs may be confusing for the reader, but if the AA ne ed all of them, the paper is sufficiently well done.

We are glad the Reviewer finds the amendments of the manuscript well done, and we thank the Reviewer for their constructive criticism.

Reviewer 2 Report

Wavelengths of light below 400 nm are not of physiological significant to the retina since they do not reach the retina.

Statements such as 'lipofuscin chromophores .. contribute.. but not to photosensizing properties' of lipofuscin are incorrect. 

THere are numerous lipofuscin fluorophores in retina that have been identified and characterized.. The oxidized forms of these fluorophores have also been characterized.  The oxidation occurs because of the photosensitizing capability of the parent fluorophores. 

The authors have not ackowledged this science

Author Response

Response to Reviewer 2:

Wavelengths of light below 400 nm are not of physiological significant to the retina since they do not reach the retina.

The visual sensitivity of the normal adult human eye extends down to 380 nm (Fundamental chromaticity diagram with physiological axes.  Parts 1 and 2. Technical Report 170-1. Vienna: Central Bureau of the Commission Internationale de l' Éclairage, 2006), which means light below 400 nm can reach the retina and evoke visual response. However, this is not relevant to our manuscript. As stated in our previous response, we used wide range of wavelengths of light allowing us to compare the absorption and other photochemical properties of oxidized docosahexaenoate with those reported for lipofuscin isolated from human eyes. We do provide data on the effects of visible light with wavelengths above 400 nm.

Statements such as 'lipofuscin chromophores .. contribute.. but not to photosensizing properties' of lipofuscin are incorrect. 

We replaced the word “indicating” with “which is consistent with the fact” in the statement in the abstract: “The quantum yields of singlet oxygen and superoxide generation by oxidized DHE photoexcited with visible light are 2.4- and 3.6-fold higher, respectively, than for lipofuscin, which is consistent with the fact that lipofuscin contains some chromophores which do contribute to the absorption of light but not so much to its photosensitizing properties.” We checked thoroughly for accuracy all other statements and confirm that they are correct.

THere are numerous lipofuscin fluorophores in retina that have been identified and characterized.. The oxidized forms of these fluorophores have also been characterized.  The oxidation occurs because of the photosensitizing capability of the parent fluorophores. 

The authors have not ackowledged this science

We have cited in the Introduction a recent review and several original papers on lipofuscin chromophores with fluorescent properties and their oxidation products, which absorb visible light (references no. 7, 17-24). Chromophores such as A2-E can undergo oxidation upon exposure to light and form oxidation products as described in the cited papers. We do acknowledge photosensitizing properties of A2-E and cite papers which report the quantum yields of generation of superoxide and singlet oxygen by A2-E (references no. 14, 28, 30, 32). These quantum yields of generation of superoxide and singlet oxygen are lower than the corresponding quantum yields for the lipophilic extract of lipofuscin. A2-E is present in lipofuscin and it does contribute to the absorption of light. This means that lipofuscin extract must contain other chromophores with quantum yields of generation of superoxide and singlet oxygen greater than the quantum yields of generation of these reactive oxygen species by the lipofuscin extract itself.